**Original Research Article**

Arabidopsis; *Brassica rapa*; comparative transcriptomics; flowering; *FT*; *SOC1*.

**Author for correspondence:**
R. J. Morris,
E-mail: richard.morris@jic.ac.uk

# Comparative transcriptomics reveals desynchronisation of gene expression during the floral transition between Arabidopsis and *Brassica rapa* cultivars

Alexander Calderwood[1], Jo Hepworth[2], Shannon Woodhouse[1],
Lorelei Bilham[2], D. Marc Jones[1,3], Eleri Tudor[2], Mubarak Ali[4], Caroline Dean[5],
Rachel Wells[2], Judith A. Irwin[2] and Richard J. Morris[1]*

[1]Department of Computational and Systems Biology, John Innes Centre, Norwich, United Kingdom; [2]Department of Crop Genetics, John Innes Centre, Norwich, United Kingdom; [3]VIB-UGent Centre for Plant Systems Biology, Gent, Belgium; [4]Bangladesh Agricultural Research Institute, Gazipur, Bangladesh; [5]Department of Cell and Developmental Biology, John Innes Centre, Norwich, United Kingdom

## Abstract

Comparative transcriptomics can be used to translate an understanding of gene regulatory networks from model systems to less studied species. Here, we use RNA-Seq to determine and compare gene expression dynamics through the floral transition in the model species *Arabidopsis thaliana* and the closely related crop *Brassica rapa*. We find that different curve registration functions are required for different genes, indicating that there is no single common 'developmental time' between Arabidopsis and *B. rapa*. A detailed comparison between Arabidopsis and *B. rapa* and between two *B. rapa* accessions reveals different modes of regulation of the key floral integrator *SOC1*, and that the floral transition in the *B. rapa* accessions is triggered by different pathways. Our study adds to the mechanistic understanding of the regulatory network of flowering time in rapid cycling *B. rapa* and highlights the importance of registration methods for the comparison of developmental gene expression data.

## 1. Introduction

During its life cycle, a plant passes through distinct growth phases, such as vegetative growth, a reproductive phase and finally seed set and senescence. These phases are separated by developmental transitions which are gated by genetic regulatory networks (GRNs).

Much of the current understanding of the genetic control of plant development arises from studies of model organisms. Comparative developmental studies, identifying similarities and differences in GRNs at equivalent timepoints, allow understanding of the model organism to be transferred to less well-studied systems, for example, from a model to crop species, or from a laboratory variety to a commercially relevant cultivar. However, different organisms develop at different rates, and one challenge in making useful comparisons is in determining equivalent timepoints.

A developing plant can be caricatured as being structurally similar to a recurrent neural network (RNN; Figure 1a). Over absolute time (as might be measured by an experimentalist's clock), the gene expression state (RNN hidden state) develops depending only on the previous timepoint's gene expression and environmental (RNN input) states. Developmental transitions are regulated by molecular machinery derived from the current gene expression state, and at each timepoint the developmental stage of the plant can be assessed based on its morphology (RNN output). In addition to absolute time, morphology (as the output of the system) is often used by experimental biologists as a means of establishing comparable timepoints, when a clear morphologically equivalent stage exists in the compared organisms. However, similarity in gene expression states (the 'hidden state') can also in principle be used as the basis of a metric to determine equivalent developmental timepoints for comparison between organisms.

Here, we explore comparative development between *Arabidopsis thaliana* and two *Brassica rapa* accessions. We use the floral transition as an exemplar genetically regulated developmental transition, due to the scientific, ecological and economic importance of flowering time, but also because the clearly defined morphological changes at the apex associated with the reproductive transition (Kinoshita et al., 2020; Tal et al., 2017) allow its timing to be accurately determined in both Arabidopsis and *B. rapa*.

In nature, flowering time is a critical factor in determining a plant's reproductive success (Ims, 1990). In agriculture, the control of flowering is important for balancing yield with developmental speed. Specifically, in north-eastern Bangladesh, demand for short-duration oilseed varieties is driven by the need to fit within a 'T. Aman rice–mustard–Boro rice' cropping pattern requiring extremely fast developing mustard varieties which can reach maturity in less than 80 days (Md et al., 2016; Miah & Mondal, 2017).

In *A. thaliana*, the transition from vegetative to inflorescence development of the apex is regulated by the complex interaction of hundreds of genes across multiple tissues (Bernier & Périlleux, 2005; Bouché et al., 2016a; Pajoro et al., 2014; Périlleux et al., 2019). These interactions comprise a GRN for flowering, which is commonly divided into a number of parallel exogenous and endogenous signalling pathways [photoperiod, ambient temperature, autonomous, vernalisation and aging (Andrés & Coupland, 2012; Bouché et al., 2016b; Hyun et al., 2019; Simpson & Dean, 2002)]. Signals from these different pathways are integrated at the apex to moderate timing of the floral transition, during which vegetative production of leaf primordia switches to production of floral primordia. This transition can be identified morphologically, and is also accompanied by changes in the expression of a number of well-characterised genes such as *FRUITFULL* (*FUL*), *SUPRESSOR OF OVEREXPRESSION OF CONSTANS* (*SOC1*), *LEAFY* (*LFY*) and *APETELA1* (*AP1*; Klepikova et al., 2015).

In Arabidopsis, exogenous signals include photoperiod and temperature, which are perceived primarily in the leaf. Under inductive environmental conditions, these signals culminate in the production of the mobile protein *FLOWERING LOCUS T* (*FT*). *FT* is able to move through the phloem to the apex where it activates flowering (Corbesier et al., 2007; Jaeger & Wigge, 2007). *FT*'s role as a signal of environmental conditions is similar in *B. rapa* (del Olmo et al., 2019). Conversely, in the perennial *Arabis alpina*, and *A. thaliana* when grown under noninductive conditions, shoots and branches can undergo the floral transition in the absence of *FT* expression, mediated by the independent endogenous aging pathway (Hyun et al., 2019).

*B. rapa* and Arabidopsis are both members of the Brassicaceae family, having diverged from their last common ancestor about 43 Mya (Beilstein et al., 2010). Given this relationship, it is likely that orthologues of the Arabidopsis genes play similar roles in *B. rapa*, and indeed, *FLOWERING LOCUS C* (*FLC*), *FT* and *SOC1* orthologues have been identified as strong candidates underlying variation in flowering time in rapid cycling *B. rapa* (Franks et al., 2015; Lou et al., 2007; Zhang et al., 2015). However, differences in the expression dynamics of floral transition genes, both between Arabidopsis and *B. rapa* and between *B. rapa* accessions, remain largely uncharacterised, and the regulatory interactions controlling the floral transition in *B. rapa* remain poorly understood (Blümel et al., 2015; Xiao et al., 2013).

Arabidopsis and *B. rapa* progress through similar developmental states based on their morphology; however, it remains unclear to what extent their progression through gene expression states is comparable. Here, we are interested in comparing this progression,

and consequently assessing (1) whether gene expression state can be used to identify equivalent developmental stages in these organisms and (2) the extent to which an understanding of the transition GRN can be transferred from Arabidopsis to *B. rapa*.

We have generated extensive transcriptomic datasets for two oilseed *B. rapa* accessions. R-o-18 is a commonly used laboratory accession, closely related to *B. rapa* oilseed crops grown in Pakistan (Rana et al., 2004). Sarisha-14 is a commercial cultivar developed at the Bangladesh Agriculture Research Institute from local varieties. It develops extremely rapidly, reaching maturity in approximately 75 days, and is thus viable in a 'rice–mustard–rice' cropping cycle (Md et al., 2016; Mia, 2017). Comparison to this unusual accession is carried out to identify commercially relevant GRN divergence in Sarisha-14 from more conventional rapid cycling oil type *B. rapa* accessions. Our dataset comprises a time course of gene expression in leaf and apex tissues for each accession, beginning during the vegetative growth and continuing through the floral transition until flower buds are visible on the plant. We compared gene expression between these varieties, and to publicly available rapid cycling Arabidopsis (Col-0) apical gene expression data (Klepikova et al., 2015).

Transcriptome comparison between Arabidopsis and *B. rapa* by alignment of gene expression profiles using curve registration (Leiboff & Hake, 2019; Ramsay & Silverman, 2005) suggests that there is not one, but many different 'developmental progressions' of gene expression running at different speeds relative to each other. There is, therefore, no single common 'developmental time' based on gene expression in these closely related plants. In addition, to identify the mechanistic basis from which these timing differences can arise, we perform a detailed comparison of differences in the gene regulatory networks controlling flowering time both between and within species. We find differences in the regulation of the apical expression of the transcription factor *SOC1* between Arabidopsis and *B. rapa*. Our data suggest an *FT*-independent mechanism for extremely rapid flowering under long-day conditions in *B. rapa* in Sarisha-14, distinct from that present in rapid flowering R-o-18.

## 2. Methods

### 2.1. Plant growth conditions, sampling, imaging and gene expression quantification

*B. rapa* cv. Sarisha-14 ($F_8$) and R-o-18 (double haploid) plants were sown in cereals mix (40% medium grade peat, 40% sterilised soil, 20% horticultural grit, 1.3 kg/m$^3$ PG mix 14-16-18 + Te base fertiliser, 1 kg/m$^3$ Osmocote Mini 16-8-11 2 mg + Te 0.02% B, wetting agent, 3 kg/m$^3$ maglime and 300 g/m$^3$ Exemptor). Material was grown in a Conviron MTPS 144 controlled environment room with Valoya NS1 LED lighting (250 µmol m$^{-2}$ s$^{-1}$) 18 °C day/15 °C night, 70% relative humidity with a 16-hr day. Sampling of Sarisha-14 and R-o-18 leaf and apex was performed 10 hr into the day. Leaf (first true leaf) and apex samples were taken over development during the vegetative growth and the floral transition, continuing until floral buds were visible (developmental stage BBCH51; Meier et al., 2009). For R-o-18, at each timepoint, in each tissue, three replicated samples were collected. For Sarisha-14, only two replicates were produced for 5 of the 17 timepoints, with three replicates of the others (see Supporting Information Table S1). Each sample consists of pooled tissue collected from three plants. Leaf and apex samples were taken from the same plants.

To identify the timing of the morphological floral transition, three to five plants were inspected under a dissecting microscope

at the same time of day as sampling for expression. The floral transition was scored when the majority of the plants scored showed domed meristems with clear round, floral primordia on the flanks (Kinoshita et al., 2020), instead of flatter meristems and flat leaf primordia (Figure 2a).

For RNA extraction, dissections were performed on ice within the growth chamber and material harvested into $LN_2$, prior to $-70\,^{\circ}C$ storage. Samples were ground in $LN_2$ to a fine powder before RNA extraction including optional DNase treatment was performed following the manufacturers standard protocol provided with the E.Z.N.A Plant RNA Kit (Omega Bio-tek Inc., Norcross, Georgia, http://omegabiotek.com/store).

For *B. rapa* accessions, 150 bp paired-end RNA reads were generated at Novogene, Beijing, China. cDNA libraries were constructed using NEB next ultradirectional library kit (New England Biolabs, Inc., Ipswich, Massachusetts), and sequencing was performed using the Illumina HiSeq X platform. An average of 60-million paired-end reads were generated per sample (Supporting Information Table S1). Publicly available gene expression data in *A. thaliana* Col-0 shoot apex from 7 to 16 days after germination grown under similar 16-hr day conditions were downloaded from NCBI SRA, project ID PRJNA268115 (Klepikova et al., 2015). Gene expression quantification was carried out using HISAT v2.0.4 (Kim et al., 2015) and StringTie v1.2.2 (Pertea et al., 2015). Reads were aligned to either the *B. rapa* Chiifu v3 reference genome (Zhang et al., 2018; R-o-18 and Sarisha-14) or the TAIR10 reference genome (Berardini et al., 2015; Col-0). Details of the alignment pipeline is given in the Supporting Information file alignment_script.sh. Gene expression level is reported in units of Trimmed Mean of M values (TMM) normalised counts (TMMC; Robinson et al., 2010; Robinson & Oshlack, 2010).

## 2.2. Comparison of gene expression states in biological samples

Orthologues of Arabidopsis genes in *B. rapa* have been previously identified in the production of the *B. rapa* Chiifu v3 reference genome, based on sequence similarity and gene synteny considerations using the SynOrthstool (Cheng et al., 2012; Zhang et al., 2018).

For comparisons of gene expression over time between Arabidopsis and R-o-18, and R-o-18 and Sarisha-14, pairs of orthologous genes with reproducible, variable expression over time were identified. For each gene in each pairwise comparison between organisms, the variance in gene expression explained by time was estimated, and genes were selected for which the variance explained was greater than 0.7 in both compared organisms. This identifies genes for which mean expression changes by a large amount between timepoints, relative to variation between replicates within each timepoint. This resulted in comparison between Arabidopsis and R-o-18 using 1,529 and 2,346 genes, respectively, and comparison between R-o-18 and Sarisha-14 using 3,097 genes.

In Figure 1c–e, gene expression distance between samples was calculated as the mean squared difference in gene expression between pairs of orthologues in Arabidopsis and R-o-18. Gene expression scaling was carried out by subtracting mean expression over the time course and dividing by the standard deviation in a genewise manner.

In Figure 4, t-Distributed StochasticNeighbour Embedding (t-SNE), was used to project all pairwisedistances between R-o-18 and Sarisha-14 samples onto one dimension, while optimally representing between sample distances (van der Maaten & Hinton, 2008). Euclidean distance between scaled gene expression was used as the distance metric. t-SNE was carried out using the 'Rtsne' function (v0.15; Krijthe, 2015), with max_iter = 16,000 and perplexity = 20.

Differential gene expression analysis between R-o-18 and Sarisha-14 vegetative apices was carried out using EdgeR (Robinson et al., 2010). Genes were filtered to only consider genes expressed >1 CPM in at least two of the six compared libraries. Genewise dispersion estimates were used during the model fitting. Gene ontology enrichment for differentially expressed genes ($p \leq .05$) was performed using clusterProfiler (Yu et al., 2012b). Details of computational environment, parameters used and so on are given in the Supporting Information file diff_expression_&_GO.R.

## 2.3. Registration of gene expression profiles over time

In order to register (align) gene expression profiles in Arabidopsis and *B. rapa*, Arabidopsis gene expression profiles over time were stretched and translated, using the least-squares criterion to determine optimality. Specifically, gene expression levels were centred and scaled using the mean and standard deviation of the overlapping registered timepoints in each species. Stretch factors of 1x, 1.5x and 2x, and translation factors between $-4$ and $+4$ days were considered. Stretching over only an arbitrary subsection of the observed time series was not considered, in order to minimise overfitting. After a candidate registration function was applied, gene expression was linearly imputed between the mean observed value at each timepoint in each species. For each gene, considered registrations were scored using the mean squared difference between *B. rapa* observed timepoint, and the imputed Arabidopsis expression value over the overlapping timepoints. The best set of registration factors for each gene minimised this score and were carried forward to compare to a no-registration model.

Bayesian model selection was used to compare the support for a no-registration model (in which expression over time for each gene is different between the two species) versus a registration model (in which expression profile differences can be resolved through the described registration procedure). For the overlappingtimepoints identified after registration, cubic spline models with six parameters were fit; to expression in each species separately ($2 \times 6 = 12$ parameters), or a single spline for gene expression in both species after the optimal 'stretch-and-translate' registration transformation had been applied ($2 + 6 = 8$ parameters). The Bayesian Information Criterion (BIC) statistic was used to compare these models for each gene.

The codes detailing the steps of gene expression registration are provided in the Supporting Information 'registration' directory.

## 2.4. Assortative mixing of registration parameters in gene interaction network

Assortativity was calculated in order to determine the extent to which genes that interact with each other share similar registration functions. Gene interactions were taken from the Arabidopsis AraNet v2 cofunctional gene interaction network (Lee et al., 2015).

FollowingNewman (2010), the assortativity coefficient was calculated as the Pearson correlation between the optimal identified registration function translation parameter among genes with have the same registration function stretch parameter, and which are directly linked in the AraNet v2 network.

Permutation testing was carried out in order to assess statistical significance. Here, identified registration stretch and translate

parameter pairs were randomly reallocated to genes, and assortativity was recalculated 100,000 times.

### 2.5. Network inference

The likelihood of regulatory links between genes was inferred from gene expression data following the Causal Structure Identification (CSI) algorithm (Penfold & Wild, 2011). The performance of this approach for data similar to ours was evaluated using synthetic gene expression data generated using networks of known structure, with varied experimental noise, correlation between candidate parents, generative GPhyperparameters and numbers of observations (Supporting Information Figure S1). Code detailing network inference is provided in the Supporting Information 'CSI' directory.

### 2.6. Identification of pri-RNA homologues

Arabidopsis and *B. rapa* precursor-mRNA sequences were downloaded from TAIR (https://www.arabidopsis.org) and miRBase (Griffiths-Jones et al., 2006). Candidate pri-mRNA gene regions were identified in the Chiifu v3 reference sequence (Zhang et al., 2018) based on BLAST similarity (E-val < 1E-20; Supporting Information Table S2). Stringtie v1.2.2 was used to reannotate the reference sequence using sequencing data from all Sarisha-14 and R-o-18 apex and leaf samples (Supporting Information file S1), and gene models overlapping the BLAST sites were considered to be candidate pri-RNA genes.

## 3. Results

### 3.1. Transcriptomes over development appear to be dissimilar between Arabidopsis and B. rapa

We compared gene expression across time, through the floral transition, in apical tissue of Arabidopsis ecotype Col-0 and *B. rapa* accession R-o-18. These closely related species move through a similar morphological sequence of developmental stages, so one might expect their transcriptomes to progress along a path of similar gene expression states. Under this assumption, we would expect to see that plants at similar morphological developmental stages exhibit similar transcriptomes (Leiboff & Hake, 2019).

To reduce noise and highlight differences and similarities in changes in developmental gene expression, we enriched the compared gene set for genes whose expression was found to change over the time course relative to variability between biological replicates (see Section 2), resulting in comparison of 2,346 *B. rapa* to 1,529 Arabidopsis homologues.

Within each species, samples taken at similar times, in general, have more similar gene expression than samples taken at dissimilar times (Figure 1c), indicating that our data are of a sufficient temporal resolution to detect developmental changes in transcriptome expression, and so identify similar developmental states. The exception to this is Col-0 Day 11, which appears highly dissimilar to all other observed timepoints, and may represent an unusually short-lived gene expression state.

To our surprise, however, no similarity can be seen in the progression of gene expression states between species (Figure 1c, upper-left and lower-right quadrants). The transcriptomes of the two species at points close in time do not appear to be more similar than the transcriptomes at more distant timepoints. This apparent lack of transcriptome similarity between organisms can be partly accounted for by differences in gene expression magnitude between organisms. After scaling gene expression in each organism (Figure 1d), later R-o-18 timepoints (from approximately 17 days) are more similar to later Col-0 timepoints (from approximately 10 days). However, the resolution at which similar stages can be seen is much less than within a species, as no developmental progression is obvious within these coarse 'early' and 'late' blocks.

Thus, despite their relatively close evolutionary relationship, apparently no similar gene expression states exist in this morphologically overlapping time course, suggesting that gene expression dynamics during the floral transition may be quite different between Arabidopsis Col-0 and *B. rapa* R-o-18.

### 3.2. Expression of key floral transition genes are similar, but differently synchronised in Arabidopsis and B. rapa

To check whether this apparent dissimilarity is due to confounding effects from genes whose expression is not involved in development, we examined the expression of key genes involved in regulation of the floral transition, and whose expression pattern is diagnostic for different developmental stages in Arabidopsis (Klepikova et al., 2015). In Arabidopsis, when *SOC1* protein expression is induced in the shoot apex, *SOC1* and *AGL24* directly activate expression of *LFY*, a floral meristem identity gene. *AP1* is activated mainly by *FT* (expressed predominantly in the leaf, so not compared here), and is also necessary to establish and maintain flower meristem identity. When *LFY* and *AP1* are expressed, flower development occurs at the shoot apical meristem according to the ABC model, through the activation of genes such as *AP3* (Lee & Lee, 2010).

Figure 2a,b shows that if only samples taken at the morphologically determined floral transition (vertical bar) are considered, expression of these key genes is similar in both species, suggesting that (as expected) these genes play a similar role in this transition in both species.

However, when expression of these five genes are considered together over time, the timing of changes in each of their expression patterns are not the same in both organisms (at least under these experimental conditions), relative to the timing of changes in the other four genes. For example, in Col-0, *SOC1* expression starts to increase before *LFY*, and plateaus prior to the floral transition, whereas in R-o-18, both genes accumulate over the same period. *AGL24* expression peaks before the floral transition in Col-0, and after it in R-o-18. In Col-0, *AP1* expression increases rapidly during the floral transition, whereas in R-o-18, it remains at a relatively low level until later in development. In Col-0, *AP3* expression increases rapidly 1 day after transition, whereas in R-o-18, there is no such increase within the first 4 days after the transition.

To study and compare the expression dynamics of these genes in more detail, we employed curve registration (see Section 2). This method aims to synchronise functional data (here, the gene expression over time of homologous pairs of genes in Arabidopsis and *B. rapa*) through the application of a suitable monotone transformation, translating and/or stretching gene expression profiles in an attempt to superimpose their dynamic behaviour.

Figure 2c shows that following registration, the expression profiles of each pair of these exemplar genes can be superimposed and, therefore, have similar (although desynchronised) dynamics in Arabidopsis and *B. rapa*. This confirms our initial expectation that the expression of homologous genes might be similar between the two species. It shows that the differences in the expression profiles of these key gene pairs are differences in the relative timing, rather than in the nature or order of expression changes.

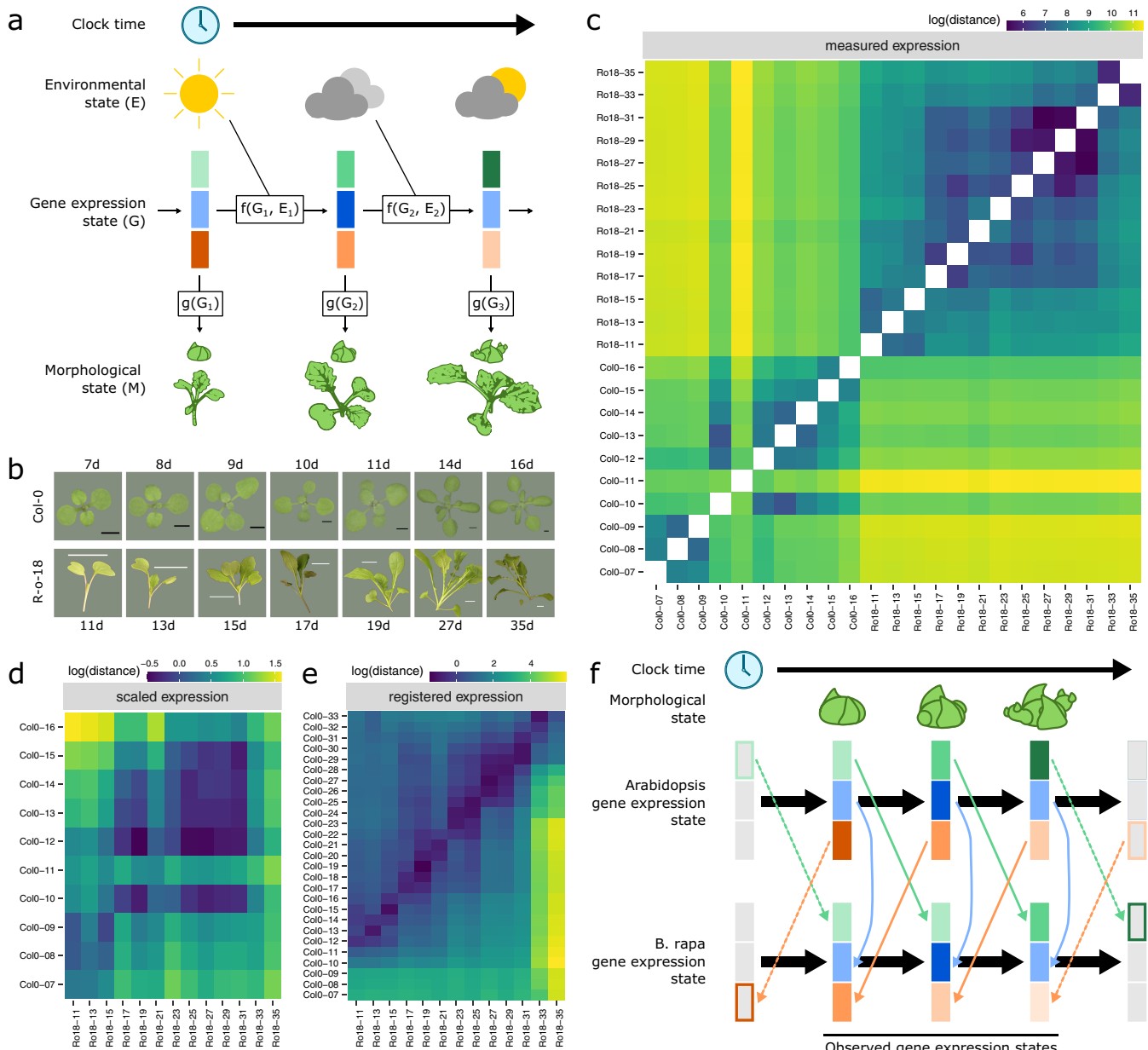

**Fig. 1.** Registration resolves differences in gene expression states during development between Arabidopsis and *Brassica rapa* in the shoot apex. (a) During its life cycle, a plant develops as a consequence of interacting environmental and gene expression states. Current developmental state is a direct consequence of gene expression and can often be assayed based on morphology. (b) Representative pictures of plants over the developmental time series. Black scale bar is 2 mm, and white scale bar is 2 cm. Col-0 images are reproduced from Klepikova et al., 2015). (c–e) Heatmaps show the gene expression distance between samples taken from the apex of R-o-18 or Col-0 at varying days after germination. Gene expression distance between pairs of samples is calculated as the average squared difference in expression between homologous pairs of genes. (c) Measured gene expression counts are not similar between species over time. For comparisons made within each genotype (lower-left and upper-right quadrants), samples taken from points close in time (points near diagonal line) are more similar to each other than to samples taken from different times (points far from diagonal). Comparing between species (upper-left and lower-right quadrants), however, reveals no obvious structure. This suggests that species in similar morphological developmental states do not necessarily exhibit similar gene expression. (d) Scaled expression values are used to control for differences in magnitude. Note the change of axes from (c) to compare only between species. In contrast to (c), some diagonal structure is now apparent, reflecting some correspondence between expression at similar times in different species. (e) Bayesian model selection suggests that for many genes, differences between Col-0 and R-o-18 are more likely to stem from desynchronisation of the same expression patterns, rather than different expression patterns per se (see Section 2). The degree of desynchronisation differs between genes, and after this is accounted for, similar gene expression states between R-o-18 and Col-0 become apparent (block structure along the diagonal). This shows that there is a common progression through more gene states than just the blocks evident in (d). (f) Genes with similar individual expression profiles exhibit different optimal registration functions between Arabidopsis and *B. rapa*, and so are differently synchronised. Here, the green gene is earlier in Arabidopsis than *B. rapa*, and the orange gene is later. Consequently, although each individual gene has a similar expression profile over time in both species, no equivalent gene expression states exist.

As can be seen in the table of optimal transformation function parameter estimates (within Figure 2c), some differences in gene expression profiles between species are found to be explained by a shift (translation) in their expression over time (e.g., *LFY*), some are found to be explained by a stretch (e.g., *SOC1*) and some require a combination of these two factors. In addition, the optimal amount of shifting and stretching differs between genes. Differences in the optimal registration function parameters of dif-

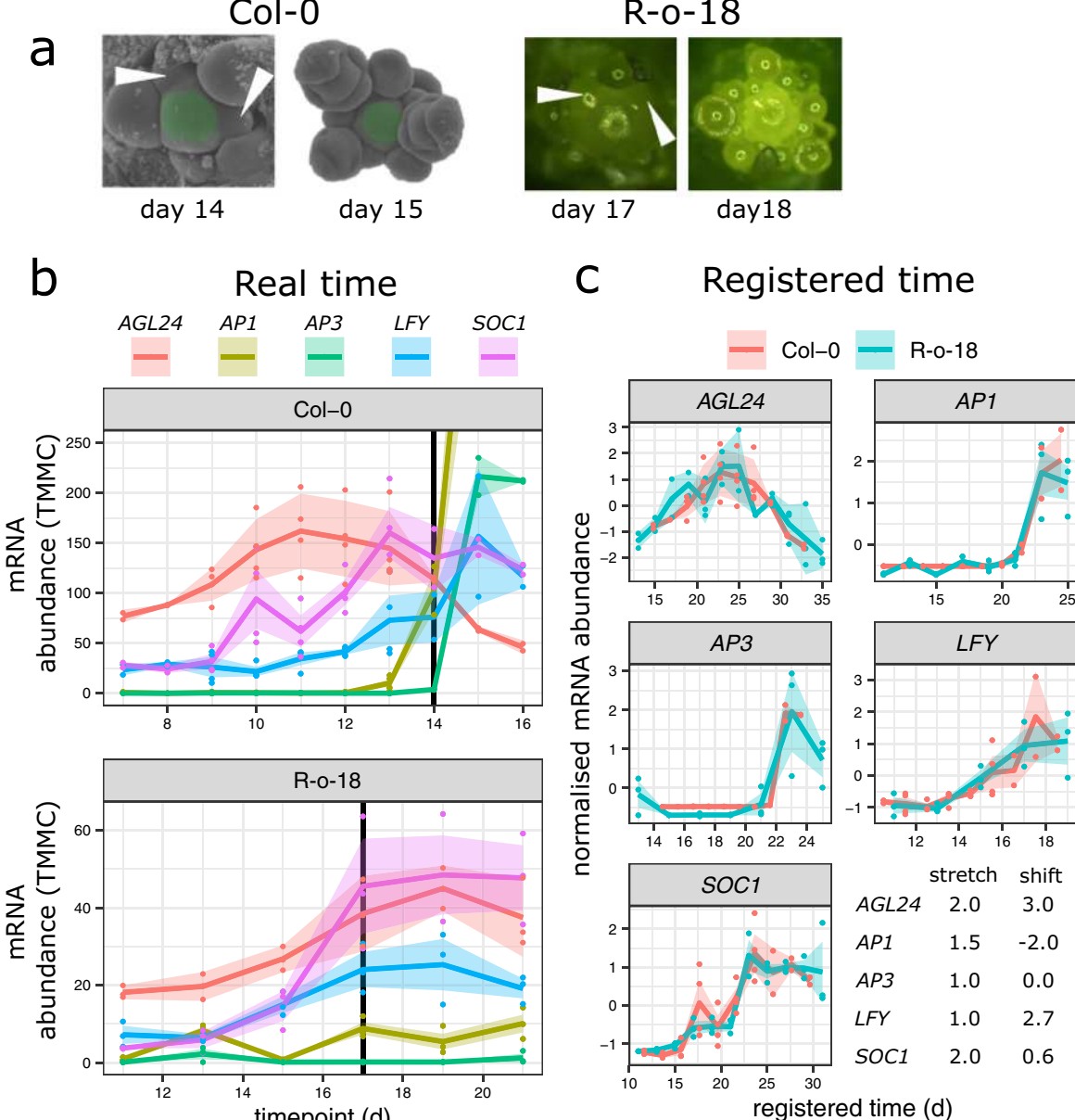

**Fig. 2.** Key floral transition genes expression profiles are similar, but their timings are different between organisms. (a) Floral transition occurs at around Day 14 in Col-0 and Day 17 in R-o-18. The earliest morphologically identifiable floral meristems are highlighted by white arrows. By the next day, the meristem is clearly floral in both cases. Col-0 SEM images are reproduced from Klepikova et al. (2015). (b) Gene expression profile for five key floral transition genes in *Arabidopsis thaliana* Col-0, and *Brassica rapa* R-o-18. Expression of paralogues in R-o-18 are summed. Morphologically identified floral transition time is identified by vertical line. The timings of gene expression changes relative to other genes, and the floral transitions differ between R-o-18 and Col-0. mRNA abundance is reported in Trimmed Mean of M values normalised counts (TMMC). (c) In spite of this, individual gene expression profiles are similar between these two organisms, as they superimpose after a registration transformation. The expression profiles of some genes are stretched out in R-o-18 relative to Arabidopsis (stretch), and also may be delayed, or brought forward relative to other genes (shift). The table shows the registration transformations applied to these genes; stretch indicates the stretch factor applied to Col-0 data and shift indicates the delay applied in days after this transformation.

ferent genes highlight that the expression patterns of these individual genes are not desynchronised by the same amount between species.

Different delays in the timing of each gene's expression means that (at least for this small set of genes) the expression of the combined set of genes is, in general, dissimilar to any single timepoint in the other species. This is the case although the expression patterns over time of the individual genes within this set are highly similar between species. When a larger set of genes (e.g., the whole transcriptome) is compared at single timepoints, these differences are likely to become more pronounced.

### 3.3. Differences in the relative timing of gene expression changes between B. rapa and Arabidopsis are common

In order to evaluate the extent to which desynchronised expression changes might explain the apparent difference in transcriptomic gene expression, we applied the same registration procedure to the full set of genes which were found to vary in expression over the time course.

We found that for 1,465 of the 2,346 considered *B. rapa* genes, the BIC favours a model that considers gene expression in *B. rapa* and *A. thaliana* to be the same (after registration) over a model in which they are considered to have different gene expression

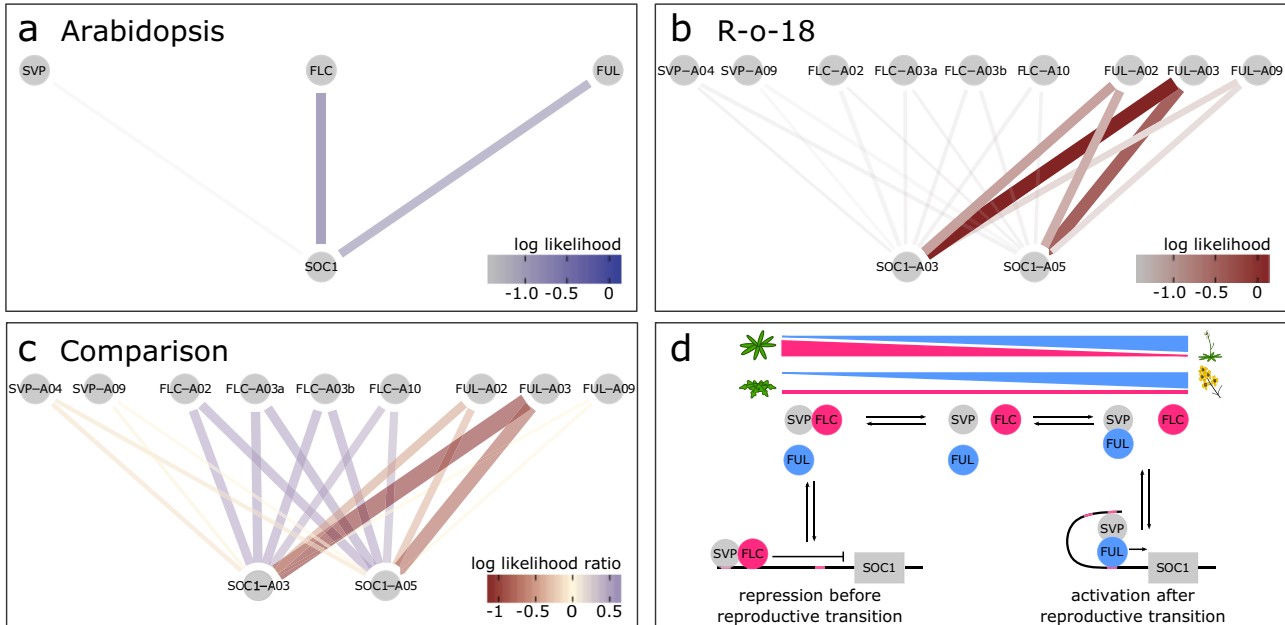

**Fig. 3.** *SOC1* is differentially regulated between *B. rapa* R-o-18 and Arabidopsis Col-0. CSI inferred gene regulatory networks between *SVP*, *FLC*, *FUL* and *SOC1* in (a) Arabidopsis and (b) R-o-18. The likelihood of the observed gene expression data given an assumed regulatory link between each pair of genes is plotted. In the absence of prior information, this is proportional to the probability of a regulatory link between the gene pair given the observed gene expression data. (c) the difference between log likelihood in Col-0 and R-o-18. Numbers after gene abbreviation indicates the chromosome numbers of the orthologue. (d) proposed mechanistic model for the role of *FUL* during the floral transition, modified from Balanzà et al. (2014), in which *FUL* and *FLC* compete to dimerise with *SVP*. In Arabidopsis, the CSI method infers that regulation of *SOC1* is via a balance of changing *FLC* and *FUL* expression. Conversely, in R-o-18, association is primarily between *SOC1*, and the A2 and A3 copies of *FUL*, suggesting that changes in the expression level of *FLC* are not relevant to controlling the upregulation of *SOC1*.

patterns. Permutation testing, in which genes in one organism are randomly allocated a comparison gene in the other, suggests that this is a significantly large number of genes to be identified ($p < 2e-23$; Supporting Information Figure S2) and, therefore, not merely an artefact of overfitting during the registration.

This analysis supports the conclusions drawn from the close examination of the few key floral genes and identifies differences in synchronisation as a general phenomenon. Thus, for many genes, the difference between R-o-18 and Arabidopsis is a delay in the gene's expression pattern, rather than a more complicated difference in their expression dynamics.

When these differences in timing are accounted for through registration, there is a further reduction in the distance between nearby timepoints and an increase in the distance between dissimilar timepoints (Figure 1e). The heatmap shows a common progression from early to late gene expression states in both species. This indicates that gene expression over time is much more similar between these organisms than could be concluded through a naïve comparison of their gene expression profiles over time without registration. It partially resolves the apparent paradox that *B. rapa* and Arabidopsis are related organisms with highly similar morphological development, but which apparently exhibit dramatically diverged gene expression patterns over development even when grown under similar environmental conditions (Figure 1f).

As in the floral gene example, different optimal registration transformation parameters are identified for different genes (Supporting Information Figure S3). Contrary to our initial hypothesis, it is, therefore, not the case that there is a single progression through transcriptomic states at different rates in *B. rapa* and Arabidopsis which could be aligned between them. Rather, there are a number of progressions bound together within each organism. These each occur at different rates, and only when they are synchronised

through different registration functions, we can see how similar they are in both species. Thus, we find that there is not, in general, an equivalent developmental stage at the transcriptome level, and therefore, no way to map both Arabidopsis and *B. rapa* to a single common developmental time in terms of overall gene expression.

### 3.4. Differently synchronised groups of genes correspond to biologically functional groups, and position in gene regulatory network

In order to identify whether known biological GRN features correspond to these differently synchronised progressions, we examined groups of genes with the same optimal registration parameters, and which, therefore, exhibit synchronised expression differences between the *B. rapa* and Arabidopsis time courses. Interestingly, groups of genes with the same optimal registration parameters are enriched in the same gene ontology terms, suggesting that they may be involved in similar functions and processes (Supporting Information Table S3). Furthermore, when superimposed over an Arabidopsis gene–gene interaction network (Lee et al., 2015), genes in the same registration parameter group are more frequently linked to each other than to genes in a different parameter group ($p < 6e-5$). Together these findings indicate that synchronised gene groups are associated with functional modules within the gene regulatory network.

That many genes have a similar expression patterns in both organisms, with cofunctional genes cosynchronised within each organism indicates that, in general, gene regulation is highly similar in Arabidopsis and *B. rapa*. It suggests that under these environmental conditions, the GRN in *B. rapa* can be usefully understood as modules of genes with highly similar regulatory relationships as in Arabidopsis (resulting in their cosynchronisation), and that

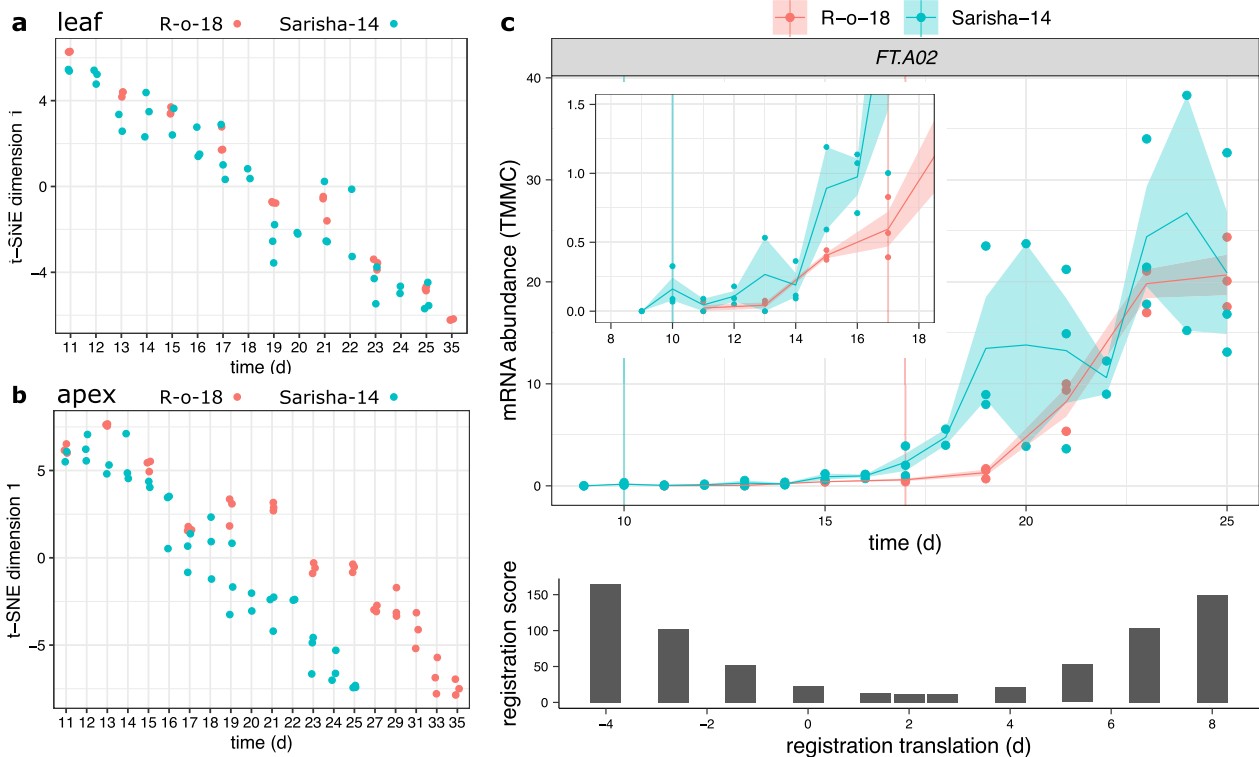

**Fig. 4.** Developmental rates differ between Sarisha-14 and R-o-18 in the apex, and is not explained by *FT* expression. Plots of time (days) against t-SNE estimated projection of gene expression to one dimension. This is an estimate of the optimal projection of the gene expression data while maintaining the correct distances between samples. Samples nearer to each other on the *y*-axis in each plot have more similar gene expression. Samples taken from (a) leaf and (b) apex in R-o-18 (red) and Sarisha-14 (blue). In leaf, development of gene expression profiles over time appears to occur at approximately the same rate between accessions, such that the most similar samples are taken at the same time. In apex, development appears to occur faster in Sarisha-14 than R-o-18. Genes were filtered to only include genes which variation over time explained >50% of variance in gene expression in both accessions. In apex, 3,097 genes were used. In leaf, 10,035 genes were used (c). Gene expression of *BraFT* in R-o-18 and Sarisha-14 over development, inset graph shows expression before Day 18, so that early gene expression behaviour can be clearly seen. Vertical lines indicate the first timepoint with floral meristems identified in each accession. mRNA abundance is reported in TMM normalised counts (TMMC). Registration indicates that expression of *FT* in the leaf is approximately 2 days advanced in Sarisha-14 relative to R-o-18. This is not sufficient to account for the 7-day difference in timing of the floral transition. Upon examination of the expression profiles, *FT* expression in the R-o-18 leaf increases between Day 13 and Day 15, prior to floral transition at Day 17. *FT* expression is not detectable in Sarisha-14 prior to the floral transition at Day 10. Expression of *FT* in the Sarisha-14 leaf at floral transition is lower than in R-o-18 (Day 17). This shows that Sarisha-14 undergoes floral transition at the apex coincident with lower *FT* expression in the leaf than in R-o-18. It is not clear from these data whether *FT* is expressed in Sarisha-14 below the experimentally detectible limit prior to the floral transition. It is, therefore, unclear from these data whether the transition occurs in response to a reduced leaf *FT* signal, or even in its absence in Sarisha-14 grown under long-day conditions.

relatively few differences in gene–gene regulatory relationships, or environmental inputs leads to desynchronisation between these modules, and differences in expression.

### 3.5. Regulation of SOC1 differs between Arabidopsis and R-o-18

To further characterise an example of a gene regulatory difference between Col-0 and R-o-18, we focus on the regulation of *SOC1* in the apical flowering time network (diploid Brassicas have three *SOC1* homologues). This transcription factor is involved in the regulation of the upstream stages of the floral transition, and (as shown in Figure 2) its expression pattern is stretched by a factor of two in *B. rapa* relative to Arabidopsis, meaning that it comes on later, relative to other genes, and is slower to progress through its expression changes. Therefore, any differences in the regulation of *SOC1* which explain this delayed expression are promising candidates to be involved in the delayed floral development in *B. rapa* relative to Arabidopsis.

To investigate potential *SOC1* regulatory changes, we derived statistical models for the GRN from the data using the Causal Structure Inference (CSI) algorithm (Penfold & Wild, 2011). Comparison of the probability of candidate gene-to-*SOC1* regulatory

links based on gene expression profiles suggests that among the largest differences in the regulation of *SOC1* between Arabidopsis and R-o-18 are changes in the response to *FLC* and *FUL* expression (Supporting Information Figure S4). Figure 3 shows that although in Arabidopsis expression of *SOC1* is consistent with regulation via repression by *FLC* and activation by *FUL* as proposed by Balanzà et al. (Balanzà et al., 2014), in R-o-18, none of the copies of *FLC* strongly associate with *SOC1*. Instead, *SOC1* expression is strongly associated with the expression of the two *FUL* paralogues located on Chromosomes A02 and A03 (BraA02G042750.3C and BraA03G043880.3C).

To understand the reason for the missing inferred regulatory links between *FLC* and *SOC1*, we considered the expression of *FLC* in more detail. Of the four paralogues of *FLC* identified in *B. rapa*, *BraFLC.A02* (BraA02g003340.3C, also called *FLC2*) and *BraFLC.A10* (BraA10g027720.3C, also called *FLC1*) have previously been reported to be nonfunctional in R-o-18 (Schiessl et al., 2017; Wu et al., 2012; Yuan et al., 2009). *BraFLC.A03b* (BraA03g015950.3C, also called *FLC5*) has previously been identified as a pseudogene, due to the deletion of exons in the reference genome (Takada et al., 2019; Wang et al., 2011). Here, we find that it is likely nonfunctional in R-o-18 as well, as it is expressed

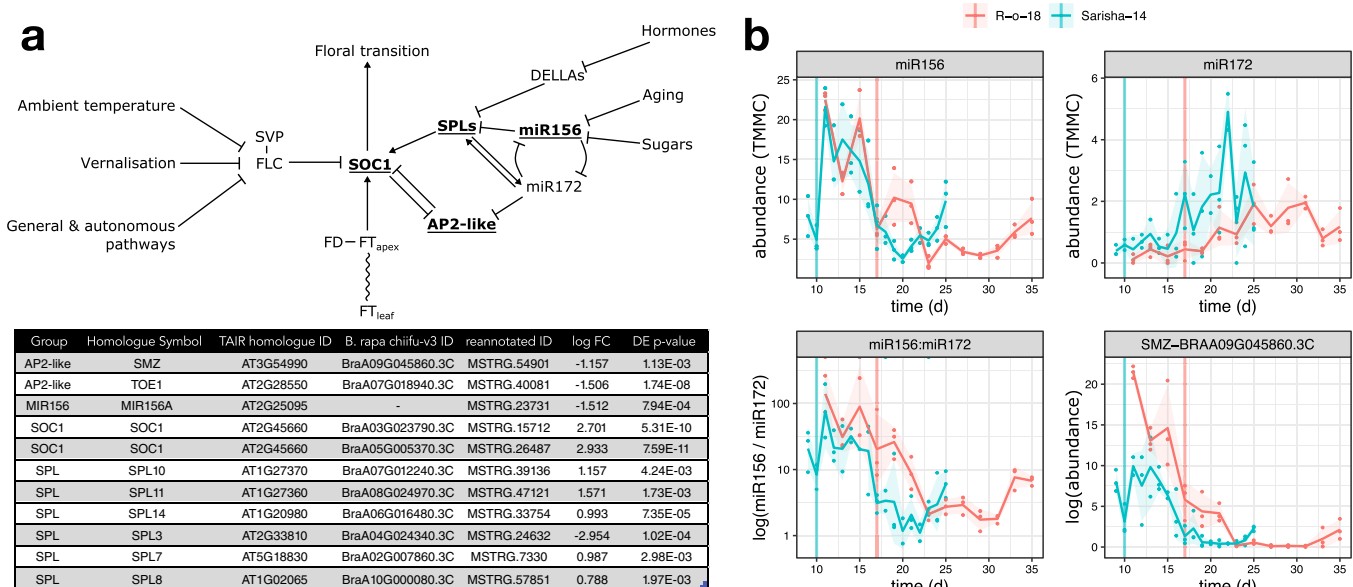

**Fig. 5.** The aging pathway proceeds more rapidly in Sarisha-14 than in R-o-18. (a) Modified from the Flowering Interactive Database website (Bouché et al., 2016b), elements which were found to be differently expressed in the apex in prefloral Sarisha-14 (Day 9) and the nearest equivalent R-o-18 sample (Day 11) are highlighted in bold and underlined. The table gives the details of differentially expressed gene identities, and log-fold change in Sarisha-14 relative to R-o-18. Differential expression of *SOC1* is coincident with differential expression of *SPLs* and AP2-like genes, rather than *FLC*, *FT*, *SVP* or *FD*, implicating the endogenous Aging, Hormone or Sugar signalling pathways in priming the early floral transition of Sarisha-14. Phytohormone signalling is integrated through the regulation of DELLA proteins. The activity of DELLA proteins is regulated posttranslationally by GA, ABA, auxin and ethylene either directly or indirectly (Achard et al., 2006; Fu & Harberd, 2003; Lorrai et al., 2018). Activities of SPLs are regulated by DELLA proteins (Conti, 2017). miR156 and miR172 are master regulators of the transition from the juvenile to adult phase of vegetative development (Wu & Poethig, 2006). During the development, initially, high levels of mature miR156 and low levels of miR172 transition to low levels of miR156 and high levels of miR172 contribute to the juvenile to adult transition (Hong & Jackson, 2015; Wu & Poethig, 2006). miR156 primarily regulates SPLs via translational regulation (He et al., 2018). *SOC1* is regulated by AP2-like transcription factors, and SPLs (Yant et al., 2010). AP2-like genes are regulated by the aging pathway, via largely via translational repression by miR172, although expression of the AP2-like gene *SMZ* has been found to depend on miR172 (Aukerman & Sakai, 2003; Chen, 2004; Yu et al., 2012a). (b) Pri-miRNA abundance is plotted as TMM normalised counts (TMMC) against days since germination. Pri-miRNA gene models were identified as described in Section 2. The ratio of miR156 to miR172 precursor RNA is lower in Sarisha-14 than in R-o-18 at equivalent timepoints. This is achieved primarily although reduced expression of pri-miR156, although pri-miR172 is also expressed at a slightly higher level in Sarisha-14 than in R-o-18. *SMZ* is transcriptionally regulated by miR172 (Yu et al., 2012a), and so its lower expression level in Sarisha-14 suggests that miR172 activity as well as precursor levels are also greater in Sarisha-14. Mean and 95% CIs are shown.

at a similar level to the other nonfunctional copies (Supporting Information Figure S5).

*BraFLC.A03a* (BraA03g004170.3C, also called *FLC3*) appears to be functional, is expressed at a higher level and does not encode a premature stop codon. In Arabidopsis, apical *FLC* expression declines prior to *SOC1* upregulation, but in R-o-18 and Sarisha-14, *BraFLC.A03a* expression declines only after *SOC1* is upregulated (Supporting Information Figure S6). This suggests a model such that in rapid cycling *B. rapa*, unlike rapid cycling Arabidopsis, the transition from vegetative to inflorescence meristem occurs prior to a decrease in expression of the floral repressor *FLC* in the apex. Consequently, the *SOC1* expression profile over development is delayed in R-o-18 relative to other flowering genes.

### 3.6. The rates of development differ between leaf and apex in B. rapa

To evaluate whether comparison of transcriptomic time series could identify variation in GRNs underlying phenotypic variation between accessions, we compared gene expression in the leaf and apex of R-o-18 and Sarisha-14 *B. rapa* varieties. R-o-18 is a well-studied yellow sarson rapid oil type. Sarisha-14 is a commercially relevant rapeseed mustard, which develops extremely rapidly, undergoing floral transition 10 days after germination and 7 days earlier than R-o-18 (Supporting Information Figure S7).

We computed the similarity in gene expression between different timepoints in R-o-18 and Sarisha-14 in leaf (Figure 4a) and apex

(Figure 4b) tissues. This suggests that in the leaf, development overall proceeds at the same rate, as the most similar samples between accessions are at roughly equivalent timepoints. This is not the case in the apex, where there again appears to be a similar developmental trajectory in terms of gene expression, but progression along this path is faster in Sarisha-14 than in R-o-18. This desynchronisation of developmental progression suggests that differences in the rate of development between these accessions likely occur at the shoot apex, rather than the leaf, and implies that differences might exist in leaf to apex signalling of the floral transition between these accessions.

### 3.7. Rapid floral transition in Sarisha-14 is not due to an early FT signal, but to increased apical sensitivity

In Arabidopsis, environmental triggers of flowering are perceived predominantly in the leaf and result in the production of *FT* protein, which moves to the apex as a component of the florigen signal (Corbesier et al., 2007; Jaeger & Wigge, 2007). This then causes upregulation of flowering genes in the apex, such as *FUL* and *SOC1* (Abe et al., 2005; McClung et al., 2016; Yoo et al., 2005). In *B. rapa*, *BraFT.A02* (BraA02G016700.3C, also called *BraA.FT.a* or *BrFt1*), which has previously been shown to be the main *FT*-like gene regulating the floral transition in R-o-18 (del Olmo et al., 2019), is the copy with the highest expression in both Sarisha-14 and R-o-18. In contrast, the *BraFT.A07* paralogue (BraA07G031650.3C, also called *BraA.FT.b* or *BrFT2*) contains a transposon insertion in R-o-

18, which is predicted to generate a loss of function allele (Zhang et al., 2015) and is not detectably expressed in either accession in our data. This suggests that it is not functional in either accession, and so is not considered here.

Meristems are floral 7 days earlier in Sarisha-14 than in R-o-18 (Supporting Information Figure S7). However, registration indicates that *FT* expression in the leaf is only approximately 2 days ahead in Sarisha-14 compared to R-o-18. We also find that CSI inferred evidence for relationships between gene expression profiles in the leaf and the floral integrator genes *FUL* and *SOC1* in the apex is weaker in Sarisha-14 than in R-o-18 (Supporting Information Figure S8). In particular, less evidence for a relationship between *FT* expression in the leaf, and changes in apical gene expression were found in Sarisha-14 than in R-o-18. Manual inspection of the expression pattern indicates that *FT* is not expressed sufficiently early in Sarisha-14 leaf relative to R-o-18 to account for the difference in the timing of the floral transition (Figure 4c). It is not detectably expressed prior to the floral transition, and at the time of floral transition, expression of *FT* in the leaf is lower in Sarisha-14 than in R-o-18 ($p = .0272$). Therefore, a given *FT* expression level in the leaf appears to result in a stronger induction of flowering response in Sarisha-14 than in R-o-18. This could be achieved by: (1) increased potency of the *FT* signalling molecule; (2) increased conductance of the signal to the apex; (3) increased sensitivity of the apex to a signal or (4) because floral transition occurs independently of *FT* in Sarisha-14. These can be considered as differences in signalling strength (Models 1 and 2) and differences in signal perception at the apex (Models 3 and 4). We find that gene expression at the apex is consistent with the third or fourth models.

To identify any differences in apical gene expression which might cause increased sensitivity to an *FT* signal, or flowering in its absence, we compared apical gene expression in the last vegetative Sarisha-14 sample (9 days after germination) and the nearest vegetative R-o-18 timepoint (11 days after germination). Both of these samples are prior to *FT* expression in the leaf, and so before differences in signal strength could affect behaviour. We found that 11,914 of 36,935 expressed genes are differentially expressed ($q < .05$), suggesting broad differences in gene expression. Among these genes, enriched representation of gene ontology terms 'positive regulation of development, heterochronic' ($q = .017$ FDR), 'shoot system morphogenesis' ($q = 2.3e-4$ FDR) and 'phyllome development' ($q = 7.7e-4$ FDR) indicate that developmental gene expression programs differ in the apex between these samples before they could be caused by *FT* signal strength differences.

We next investigated whether differences in other signalling pathways in the apex could account for the apparently different *FT* signal sensitivity. In Arabidopsis, the floral transition is controlled by multiple interacting pathways that are sensitive to environmental cues, as well as developmental age, which is controlled by a complex interaction between phytohormone signalling, sugar status and the activity of microRNAs miR156 and miR172, and prevents premature flowering in juvenile plants. Signals from these different pathways are perceived and integrated at the shoot apex (Figure 5a). We identified differently expressed genes in the miR156-SPL and AP2-like regulatory modules, but not in expression of *FLC*, *SVP* or *FD*. In particular, we note that that expression of miR156, miR172 and *SCHLAFMÜTZE* [*SMZ*; the only AP2-like gene which is found to vary in transcriptional expression in response to perturbed miR156–miR172 expression (Yu et al., 2012a)] are similar in Sarisha-14, and R-o-18 immediately prior to the floral transition

in both accessions (Figure 5b), although these events occur 1 week apart in time from germination.

Therefore, differences exist in the apical expression of key components of the endogenous developmental age pathway. This signalling pathway interacts with *FT* signals from the leaf to regulate the floral transition. These findings are consistent with a model in which the early floral transition in Sarisha-14 is caused primarily by differences from R-o-18 in *FT* signal sensitivity at the apex (Model 3 or 4), mediated by priming via other signalling pathways, rather than due to differences in *FT* signal generation in the leaf.

## 4. Discussion

Research into the mechanisms of regulation of the floral transition has focussed largely on the model organism Arabidopsis. This has generated a demand for methods for translating this knowledge to other species. Here, we demonstrate that apparently large differences in gene expression profiles over development between the closely related crop *B. rapa* and Arabidopsis can mostly be resolved through the application of a curve registration step during the data analysis. We found that different genes require different registration functions, consistent with the desynchronisation of multiple regulatory modules within the GRN between these species. We identified exemplar differences in the regulation of the floral integrator gene *SOC1* between rapid cycling Arabidopsis and *B. rapa* in these developmental time courses. Through comparison of gene expression profiles in R-o-18 and Sarisha-14, we have identified a putative *FT*-independent mechanism which potentiates the extremely early floral transition in Sarisha-14 and consequently underlies its commercial viability in Bangladesh.

### 4.1. Registration of gene expression states between Col-0 and R-o-18

Arabidopsis and *B. rapa* are evolutionarily related and exhibit similar transitions in apex morphology over development. We were, therefore, surprised that equivalent gene expression states do not apparently exist in apex tissue taken from these organisms over development.

Registration of gene expression profiles shows that this difference is partly caused by desynchronisation of gene expression between organisms, and that gene expression profiles are, in fact, similar when considered on an individual gene basis. Furthermore, genes that require similar registration functions for alignment between species appear to be involved in the similar biological processes.

This suggests that rather than the single gene expression state progression (as shown in Figure 1a), the gene expression layer can be decomposed into a number of progressions (with different genes in each), each relating to the state of different biological developmental processes, and with limited regulatory crosstalk between them.

Consequently, it appears that for the purposes of comparison between species, development, in general (and not just development as defined by gene expression state), is multidimensional. Progress in different developmental processes occurs at different rates between species, and therefore, even between these closely related species, developmental comparisons which project this back to one-dimension (such as comparisons based on morphology, or a simple mapping between most similar general gene expression states using all genes) may risk oversimplification.

Although this means that there is no experimentally simple way to compare developmental time between species, the apparent existence of shared gene expression progressions does suggest that some relatively low-dimensional 'developmental manifold' in gene expression space exists, and that samples from different species can be defined in terms of their position within it. However, it seems likely that for the purposes of developmental comparison, the correct timepoints to be compared will depend on the biological process of interest, rather than being universally equivalent.

It will be interesting to determine the extent to which consistently synchronised gene groups exist between additional species, and how closely these cofunctional gene groups relate to manually annotated functional ontologies.

### 4.2. Flowering GRN in Col-0 versus R-o-18

Comparison of optimal registration functions for expression profiles of key floral genes indicates that expression of *SOC1* is delayed in *B. rapa* versus Arabidopsis relative to other gene expression profiles under these environmental conditions. Detailed comparison of patterns between gene expression profiles using the CSI algorithm identified differences in the relationships between expression of *FLC* and *FUL*, and *SOC1*. In Arabidopsis, *SOC1* is partly regulated by the balance of *FLC* and *FUL*, which compete to dimerise with *SVP*. *FLC–SVP* represses *SOC1* expression, whereas the *FUL–SVP* dimer activates it (Balanzà et al., 2014). Over time, apical *FLC* expression declines and *FUL* expression increases to the point that *FUL–SVP* becomes the dominant dimer. Gene regulatory links inferred from Arabidopsis gene expression data are consistent with this model; however, those from *B. rapa* are not. Instead, in *B. rapa*, *BraFLC* expression remains high until after *BraSOC1* expression is well established, although it appears to encode a functional protein. This suggests that a different mechanism for the regulation of *SOC1* expression exists under these conditions in R-o-18.

R-o-18 is commonly used as a model Brassica accession due to its rapid life cycle and lack of vernalisation requirement, yet this analysis suggests that it could potentially be made to flower more rapidly. An interesting breeding objective to achieve this end would be to knock out expression of the *BraFLC.A03a* copy in the apex. We hypothesise that this may reduce competition for *SVP* dimerisation, and allow precocious upregulation of *SOC1* expression, and subsequent changes in the regulation of its downstream target genes.

### 4.3. Flowering time in Sarisha-14 versus R-o-18

In Arabidopsis, flowering can be triggered under long-day, inductive conditions by *FT*, or by aging and phytohormones under noninductive, short-day conditions (Hyun et al., 2016; 2019). Differences in the timing of the floral transition between R-o-18 and Sarisha-14 are not accounted for by differences in the expression profiles of *FT* homologues, which have similar expression patterns and levels until much later in development. Many components of the aging pathway to floral transition are under post-transcriptional control (Figure 5a), and so not directly identifiable by RNA-seq. It is striking that gene expression of those key components which can be detected are consistent with differences in this pathway between Sarisha-14 and R-o-18.

Previous studies have identified a transposon insertion in the second intron of R-o-18 *BraFT.A7* which causes a reduction of expression as underlying a QTL between R-o-18 and the fast flowering Caixin-type L58 (Zhang et al., 2015). However, while in

L58, similar expression levels were observed for both copies of *BraFT*, in Sarisha-14, we observed the same reduced expression of *BraFT.A7* as in R-o-18, indicating that this allele does not underlie the difference between Sarisha-14 and R-o-18.

*FT* expression is known to vary over the course of a day in Arabidopsis (Krzymuski et al., 2015; Song et al., 2018). Although samples from both varieties were taken at the same time, it is possible that differences in the expression dynamics over the diurnal cycle contribute to differences in development. It is also possible that potential differences in *FT* signalling effectiveness, or in tissue conductivity to long distance signals, contribute to differences in *FT* activity at the apex, which cannot be seen in gene expression level in the leaf. However, we see no evidence for differences in the *FT* coding sequence between Sarisha-14 and R-o-18, and we do see evidence for differences in phytohormone and age-related signalling. Consequently, differences in the GRN at the apex is the most parsimonious explanation for the early flowering phenotype.

Interestingly, selective breeding appears to have produced a variety, Sarisha-14, that uses the aging GRN to trigger early flowering. The aging GRN can be viewed as an endogenous timer that normally acts in older meristems to allow flowering in the absence of *FT* under unfavourable environmental conditions (Hyun et al., 2019). In Sarisha-14, however, it apparently proceeds so rapidly that it becomes a trigger for flowering even under inductive, long-day environmental conditions, either in the absence of *FT*, or under lower concentration than is required in R-o-18.

A challenge in determining the causal genomic differences between R-o-18 and Sarisha-14 is that the identified GRN is highly connected, incorporating post-transcriptional regulation and many key developmental phytohormones and sugar signalling into the regulation of aging. Identifying the causal alleles will, therefore, likely require use of a recombinant inbred line population.

## 5. Conclusions

Flowering time control is of major importance in crop adaptation to different environments. Our study provides gene expression data for all genes in leaf and apex for two rapid cycling oil type *B. rapa* lines through the floral transition. By curve registration of gene expression profiles, and network inference, we have identified differences in the regulation of the floral transition between Arabidopsis and *B. rapa*. We also identified regulatory differences between *B. rapa* varieties and linked these to phenotypic differences. This demonstrates that GRNs differ even between closely related cultivars. The data presented provide a foundation for future breeding efforts of *B. rapa* crops.

## Acknowledgements

We are grateful to Dr Lei Zhang for providing an updated identification of paralogous *B. rapa* genes and mapping to Arabidopsis homologues. We thank Profs Lars Ostergaard and Steve Penfield for critical reading, valuable comments and suggestions.

**Financial support.** The authors acknowledge financial support from the Global Challenges Research Fund grant 'Optimising maturity for enhanced yield in the Sarisha crop, Bangladesh' (BB/P01531X/1), and the Biotechnology and Biological Sciences Research Council grants 'Brassica rapeseed and vegetable optimisation (BB/P003095/1) and 'Genes in the environment' (BB/P013511/1).

**Conflict of interest.** The authors declare no significant competing financial, professional or personal interests which might influence the performance or presentation of this study.

**Authorship contributions.** A.C. designed and performed the majority of the data analysis. D.M.J. provided technical support. J.H., E.T., S.W., L.B., A.C., J.I. and R.W. performed the experiments. M.A. and C.D. provided experimental material. R.J.M., R.W. and J.I. supervised the project. A.C. planned and wrote the first draft of the manuscript. A.C., R.W., J.I., J.H. and R.J.M. wrote the manuscript with contributions from all authors. All authors provided critical feedback that helped shape the analysis and the manuscript.

**Data availability statement.** The Illumina sequence reads have been deposited in NCBI Sequence Read Archive, project ID PRJNA593493. Scripts are made freely available as supporting information.

**Supplementary Materials.** To view supplementary material for this article, please visit https://dx.doi.org/10.1017/qpb.2021.6.

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
