## [Reviewer Report]

*Comments to Author*: Calderwood and collages perform a comparative transcriptomic analysis of floral meristems between two Brassica rapa cultivars and the model plant Arabidopsis thaliana. They provide new B. rapa RNA-seq data in addition with a detailed gene expression comparison of the different RNA-seq dataset. They conclude that there are differences between Arabidopsis and B. rapa, and highlight alternative modes of flowering time regulation between these species and among rapid cycling B. rapa cultivars.

I found the work very interesting but there are several issues about the methodology and the conclusions that require clarification.

Issues of major concern

1. In the Methods (line 145-147), it is not clear if there are replicates samples from each time point. In addition, the number of pooled apex per sample, three, is rather short considering the variability in expression among different plant individuals.

2. I could not find information about number of total sequenced reads for each sample, mapping statistics, sequence quality data, etc. 

3. The bioinformatic methods are described in full, in most cases there is not detailed setting description. To allow reproducibility the authors should provide more detailed protocols, maybe as supporting information, of the performed analyses. 

4. Line 173, how the Arabidopsis and B. rapa homologs were identified it is not explained in the text.

In the Results, the first sections (pages 10 to 13) deal with the comparison of the gene expression profile, through the floral transition, in apical tissue of Arabidopsis and B. rapa accession R-o-18. After a number gene expression profile registration analyses the authors conclude that “there is no common developmental time between Arabidopsis and B. rapa”. Although I see the point of the authors and it is clear that there are different dynamics of key floral regulators in different species I have some concerns about how the time of the floral transition was scored by the authors.

5. How the authors determined the morphological changes associated with floral transition in the biological samples used for the transcriptomic analysis? There are some pictures of the comparison between R-o-18 and Sarisha-14 (Fig.S7), are these pictures from the same set of plants used in the RNA-seq experiment?

6. In Fig.2a, there is a vertical line indicating the morphological changes in the meristem associated with the floral transition. I found surprising that floral transition happens at 14 days in Arabidopsis and at 17 days in B. rapa R-o-18, because in my opinion the life span of both species it is not so close. In fact, the expression data showing that B. rapa AP1 and AP3 homologs genes are no up-regulated (Fig.2) at 17 days suggest to me that 17 days could be earlier that the floral transition in B. rapa.

7. In the results page 14-15, I found very interesting the data the differential regulation of SOC1 expression. However, I found redundant the model in Fig3d because it is not very clear and it only describes Arabidopsis knowledge. I would rather move some gene expression data from FigS6 to the main body of the manuscript than having that model.

8. About the Discussion (pages 18 to 20), there is almost nothing about the first part of the results – the curve registration analyses, etc (pages 10 to 13). Most of the Discussion is about SOC1 and the differences between R-o-18 and Sarisha-14. I suggest the authors to balance the Results and Discussion sections accordingly. 

Minor points: 

9. Line 167, I think that there are not “hundreds of genes” regulating the floral transition, maybe the authors meant that “hundreds of genes” change their expression during this developmental transition.

10. The standard nomenclature of Brassica genes is sometimes confusing. The different FLC (FLC1, FLC3, etc) and FT (BraA.FT.a) homologs have been named in previous publications. The authors could include these names together with theirs to aid the reader.

---

## [Reviewer Report]

*Comments to Author*: Calderwood et al., study a very frequent conundrum for developmental biologists : shall I compare mutant and WT at similar age or similar stage? Here they use flowering time in B. rapa cultivars and Arabidopsis as a case study. Building on a valuable RNAseq dataset, they use a registration method to compare the GRN dynamics, and find that stretching time can indeed reveal common principles as well as structural differences. This is a major achievement. The authors go one step further by also dissecting some of the differences in the GRN. One could think that for so closely related species, the flowering time GRN obtained in Arabidopsis would translate to other species without too much deviation. They instead find significant differences. This is a very interesting quantitative work, with important take-home messages. However in its current state, it may appear too much as a specific flowering time paper, see my suggestions below.

1/ I’m wondering whether the authors could try to take one more step back, and use their study to derive a more generic message. A little bit like what I tried to do in this opening (i.e. putting more emphasis on registration). They might also want to consider the possibility that time (and sensitivity) become GRN components, in a relativistic framework (a bit as in Jaeger 2008 Development doi:10.1242/dev.018697). This could be done in the abstract, introduction and discussion, and with a general self-explanatory graphical abstract (thus not necessarily on flowering time – it could be a panel in figure 1 to put the question on the table)

2/ From this study, can we conclude that the GRN from the fast cycling Arabidopsis is biasing our understanding of flowering time control? I think the authors show something that echoes a little bit what O. Leyser et al. found when they discovered that Arabidopsis was one of the only plant species with only 1 PIN1 gene. Other have shown how peculiar the Arabidopsis genome is, and the authors could discuss this point further, expanding from flowering time to other functions with different GRNs in other plant species.

3/ Figures are not always self-explanatory:

- Figure 1 should include a picture of the real plant material.

- Figure 2: next to “a”, write “real time”, next to “b” write “registered time”. A sketch of the plant phenotype (only the non-registered time) could be added on panel “a” to be crystal clear. CPM (number of reads) should be explained somewhere in the manuscript (actually I’m not sure “expression” is OK for the y axis, it should rather be mRNA accumulation).

- Figure 3: Add some text next to a, b,… to facilitate the reading of the figure without the need to go to figure caption. Minimally, add (a) Arabidopsis, (b) R-o-18.

- Figure 4 is obviously missing a color code on the figure panel and a title for each panel

- Figure 5 : I don’t understand why there is a discrepancy between the inset (before day 18) and the main panel which includes the time window in the inset, but with a different y axis. Is one of them registered ? This is unclear.

- Figure 7 is supposed to show differences in sensitivity, but this is not graphically rendered. Maybe an introductory panel would help.

4/ An extra minor point: « In agriculture, the control of flowering is important for determining yield, and must be optimised to fit within the constraints of the growing season. » I always trouble with the word « must » (deterministic) and « optimization » (since it may imply that constraints are known). I would simply say « and usually fit with the constraints… »

---

## [Reviewer Report]

*Comments to Author*: We have now received two reviews, which find the work very interesting contribution to the field. Both reviewers, however, request clarifications and adaptation regarding methodology and presentation of the work.

---

## [Reviewer Report]

Dear QPB, 

We thank the editor and reviewers for their supportive comments and constructive critique. We think the revised manuscript has been significantly strengthened by following the recommended changes and addressing the questions and comments raised by the reviewers.

We hope the current version is clearer and we look forward to receiving feedback.

With best wishes

Richard

---

## [Reviewer Report]

*Comments to Author*: I found that this new version is really improved and I appreciate the effort of including detailed bioinformatic methods. Overall I am happy with all the changes and answers.

However, I have one minor comment: I think it should be mention in the main text that the are several B. rapa SOC1 homologs. This fact may not be clear for the average reader. Maybe you could add a sentence somewhere after line 511 (Regulation of SOC1 differs between Arabidopsis and R-o-18).

---

## [Reviewer Report]

*Comments to Author*: The authors have addressed all my comments. This is a great paper!

Maybe one last suggestion. Iߣm wondering whether the following title would be more provocative: "Comparative transcriptomics reveals desynchronisation of gene expression during floral transition between Arabidopsis and Brassica rapa cultivars"

---

## [Reviewer Report]

*Comments to Author*: Dear Richard,

we have now received back the comments from the reviewers and they find the manuscript greatly improved and ready for publication. However, they have one suggestion regarding the title and one regarding B. rapa SOC1 homologs, which you may want to consider. 

Congratulation to this nice paper,

Christian

---

## [Reviewer Report]

Dear QPB, 

We have addressed the remaining issues with our manuscript and think it has been greatly improved by the review process. 

We you and the reviewers for your constructive feedback. 

On behalf of all authors, 

Richard Morris